# Three-Dimensional Unsteady Axisymmetric Viscous Beltrami Vortex Solutions to the Navier–Stokes Equations

**Koichi Takahashi**

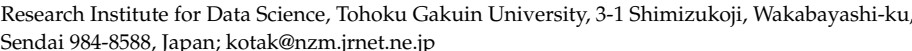

Research Institute for Data Science, Tohoku Gakuin University, 3-1 Shimizukoji, Wakabayashi-ku, Sendai 984-8588, Japan; kotak@nzm.jrnet.ne.jp

**Abstract:** This paper is aimed at eliciting consistency conditions for the existence of unsteady incompressible axisymmetric swirling viscous Beltrami vortices and explicitly constructing solutions that obey the conditions as well as the Navier–Stokes equations. By Beltrami flow, it is meant that vorticity, i.e., the curl of velocity, is proportional to velocity at any local point in space and time. The consistency conditions are derived for the proportionality coefficient, the velocity field and external force. The coefficient, whose dimension is of [length$^{-1}$], is either constant or nonconstant. In the former case, the well-known exact nondivergent three-dimensional unsteady vortex solutions are obtained by solving the evolution equations for the stream function directly. In the latter case, the consistency conditions are given by nonlinear equations of the stream function, one of which corresponds to the Bragg–Hawthorne equation for steady inviscid flow. Solutions of a novel type are found by numerically solving the nonlinear constraint equation at a fixed time. Time dependence is recovered by taking advantage of the linearity of the evolution equation of the stream function. The proportionality coefficient is found to depend on space and time. A phenomenon of partial restoration of the broken scaling invariance is observed at short distances.

**Keywords:** hydrodynamics; incompressible fluid; vortex solution; Beltrami flow; stream function; scaling invariance; Bragg–Hawthorne equation





## 1. Introduction

Three-dimensional swirling vortex solutions to the Navier–Stokes equations have been of researchers' particular interest because of their immediate relevance to phenomena in nature, laboratory and practical engineering. Besides numerical solutions, exact analytical solutions should make it possible to deepen the understanding of the details of the structure of vortices characterized by speed, vorticity, pressure, shearing, stretching, stagnation, etc.

In solving the non-linear Navier–Stokes equations, it is a common practice to assume the functional form of a solution that reduces the original partial differential equations to a more tractable system of ordinary differential equations. The well-known vortex solutions found in this way involve the steady axisymmetric one-cell solution by Burgers [1] and Rott [2], the two-cell solution by Sullivan [3], and the unsteady multi-cell solutions by Bellamy-Knights [4,5]. Solutions with a line singularity were presented in [6,7]. Solutions which temporally approach the Burgers' solution were studied in [8].

It was not clear if the unsteady solutions in [4,5] were linked to the steady solutions in [1–3]. Recently, these steady and unsteady solutions were treated in a unified way in [9], wherein the "instability" of these solutions were shown to be attributed to their extreme sensitivity to boundary conditions. For other vortex solutions and a review, see Drazin and Riley [10].

All the solutions given above are similarity solutions that have been obtained by taking advantage of the invariance of the Navier–Stokes equations under the scaling transformation. As a consequence, some components of their velocity fields linearly diverge at spatial infinity. Hence, the solutions are regarded as modeling flow patterns near the symmetry axis or plane.

Observable physical excitations on a uniform background flow must have a finite energy density, except for logarithmic singularity [9]. In particular, the divergence at spatial infinity will be problematic when we are interested in investigating the global nature of a vortex. The primary interest in this paper is to find three-dimensional unsteady viscous swirling vortex solutions whose energy density is globally not divergent. (In this paper, by "three-dimensional", we mean that there is no Cartesian coordinate system in which one of the velocity components identically vanishes).

Beltrami flow, whose vorticity is parallel to velocity in the whole space under consideration, is expected to be a promising candidate for the above purpose because of the existence of the proportionality coefficient, a scalar quantity of dimension $[\text{length}^{-1}]$, thereby raising the possibility of generating vortices of finite extension in space and/or time. Incidentally, we recall that tilting the vorticity is also accompanied by the breaking of the scaling invariance [8].

The marked characteristics of the Beltrami flow, the study of which was originated by Trkal [11] and Gromeka [12], is that the vorticity equations are apparently linear by virtue of the vanishing vector product of vorticity and velocity. In particular, inviscid Beltrami flows obeying the Euler equations have been of researchers' intense concern because of their close relevance to the flows with large Reynolds numbers that are expected to model the flows in meteorology [13,14] or engineering [15–17]. The so-called ABC flow that shows rich physical content in the Lagrangian dynamics provides a key to understanding chaotic or dynamo phenomena in fluids [18–21].

Steady axisymmetric inviscid Beltramian solutions in bounded or unbounded space have been found by focusing on the so-called Bragg–Hawthorne equation [17,22,23]. How the structure of the fundamental modes admitted by the linear equation varies by depending on the coordinate system chosen for solving the equation has been revealed in [22,24]. Non-axisymmetric solutions with vanishing helicity, i.e., inner products of velocity and vorticity, as well as the vortices with a constant proportionality coefficient, $c$ hereafter, are presented in [24]. A Beltrami flow whose $c$ is constant is called a Trkalian flow.

Despite the seemingly quite artificial setting for the Beltrami flows, the physical mechanism of generating a Beltramian (i.e., force free magnetic) field was considered as early as about 70 years ago by Chandrasekhar [25] and Chandrasekhar and Woltjer [26] in an astrophysical context with a suggestion of a principle of minimum dissipation and maximum energy. Let us recall that, with the correspondence of magnetic field to vorticity, the force-free motion of charged particles in magnetic field has mathematical similarity to Beltrami flow. The idea in [25,26] parallels the expectation in the Newtonian fluid dynamics that an increase in the Reynolds number tends to align velocity and vorticity [27,28]. Later, the importance of the maximization of helicity [21] and the minimization of magnetic energy due to dissipation in the evolution of magnetic field of nontrivial topology has been pointed out in [29,30]. It is convincing to explore the role of Beltrami flows in meteorology [31] and plasma physics [32,33] where strong flows participate. There exist profound physical meanings in studying Beltrami flows.

It is known that a steady incompressible viscous Beltrami flow is possible when external force is nonconservative. Steady compressible solutions were presented in [34]. Examples of unsteady Beltrami flow with a constant $c$ were given by [11,35]. That three-dimensional unsteady incompressible viscous Beltrami flows with a constant $c$ decay exponentially in time is also argued by [36,37]. These solutions are specified by, in addition to $c$, a wavevector that describes spatial oscillations of the flow. In this paper, we shall call such a flow (solution) a single mode flow (solution).

The works by [11,35–37] assert that the viscous Trkalian velocity field of a single mode in an unbounded space exponentially decays in time with the characteristic time proportional to $1/\nu c$ with $\nu$ being the kinematic viscosity. The appearance of this time scale is due to the parameter $c$ of the dimension $[\text{length}^{-1}]$, thereby violating the scaling invariance of the original Navier–Stokes equations. With a characteristic length scale being defined, it is thus expected that the spatial extension of vortex solution may also

be rendered finite. The generalized Beltrami flows [10,38,39] and the extended Beltrami flows [40,41] will share the same features in common.

An axisymmetric genuine meridional Beltrami vortex does not exist (see the next section). Local alignments of velocity and vorticity, meanwhile, are observed in numerical simulations of turbulence [27,42–44]. A constraint on the time evolution of a spatially periodic flow from an arbitrary initial condition was given in [45]. That the incompressible Beltrami flow decays timewise exponentially if *c* does not depend on time was shown in [46]. These works naturally let us nourish an expectation that the Beltrami flows with the spatially variable proportionality coefficient exist. If that is the case, recalling that the decay time is $1/\nu c$, we may address a question: how is the anticipated "rapid" decay of the flow avoided especially for small Reynolds numbers? To the author's knowledge, any works on this problem have not been reported thus far.

Our plan in this paper is as follows. We derive and solve the evolution equation and the constraint equation for the stream function and the coefficient *c* on the projected meridional motion, i.e., a projection of the three-dimensional flow onto the local meridian plane in the cylindrical coordinates. The derived constraint equation corresponds to the Bragg–Hawthorne equation that has been frequently used for studying steady inviscid flows. However, our constraint does not depend on viscosity. An obstacle is that the constraint equation, although it reduces to the Bragg–Hawthorne equation in a special case, is generally nonlinear and is difficult to be solved analytically in the whole space and time. However, since the evolution equation is linear and the temporal behaviors of single mode solutions are uniquely determined, we can construct the entire time-dependence by superposing the single mode solutions obtained by mode-decomposition of a numerical solution of the constraint equation. Of course, it must be assured that the superposed flow is Beltramian.

The outline of the paper is as follows. After the implication of the axisymmetric viscous Beltrami vortex is reviewed and summarized, we derive the constraint equation for unsteady three-dimensional axisymmetric viscous swirling Beltrami vortices. It is proved that, if the velocity field is time-dependent and *c* is variable, *c* is time-dependent, too. Finally, the well-known exact vortex solution for constant *c* and a new solution with nonconstant *c* are constructed.

## 2. Dynamical and Constraint Equations for the Axisymmetric Beltrami Vortex

The Navier–Stokes equations for the respective components of the velocity field in cylindrical coordinate $(r, \theta, z)$ are expressed as

$$\frac{\partial v_r}{\partial t} + v_r \frac{\partial v_r}{\partial r} + \frac{v_\theta}{r} \frac{\partial v_r}{\partial \theta} + v_z \frac{\partial v_r}{\partial z} - \frac{v_\theta^2}{r} = \nu \left( \nabla^2 v_r - \frac{v_r}{r^2} - \frac{2}{r^2} \frac{\partial v_\theta}{\partial \theta} \right) - \frac{1}{\rho} \frac{\partial p}{\partial r} + F_r \quad (1)$$

$$\frac{\partial v_\theta}{\partial t} + v_r \frac{\partial v_\theta}{\partial r} + \frac{v_\theta}{r} \frac{\partial v_\theta}{\partial \theta} + v_z \frac{\partial v_\theta}{\partial z} + \frac{v_r v_\theta}{r} = \nu \left( \nabla^2 v_\theta - \frac{v_\theta}{r^2} + \frac{2}{r^2} \frac{\partial v_r}{\partial \theta} \right) - \frac{1}{\rho} \frac{1}{r} \frac{\partial p}{\partial \theta} + F_\theta \quad (2)$$

$$\frac{\partial v_z}{\partial t} + v_r \frac{\partial v_z}{\partial r} + \frac{v_\theta}{r} \frac{\partial v_z}{\partial \theta} + v_z \frac{\partial v_z}{\partial z} = \nu \nabla^2 v_z - \frac{1}{\rho} \frac{\partial p}{\partial z} + F_z \quad (3)$$

where $v = (v_r, v_\theta, v_z)^T$ is the velocity, $\nu$ the kinematic viscosity, $p$ the pressure, $\rho$ the density, and $\boldsymbol{F} = (F_r, F_\theta, F_z)^T$ the external force. In this paper, we restrict ourselves to axisymmetric flows so that all the $\theta$-derivative terms in (1)~(3) are ignored.

A Beltrami flow is one whose velocity and vorticity are parallel:

$$\boldsymbol{\omega} = \begin{pmatrix} \omega_r \\ \omega_\theta \\ \omega_z \end{pmatrix} = \begin{pmatrix} -\partial_z v_\theta \\ \partial_z v_r - \partial_r v_z \\ r^{-1} \partial_r (r v_\theta) \end{pmatrix} = c \begin{pmatrix} v_r \\ v_\theta \\ v_z \end{pmatrix} = c\boldsymbol{v}, \quad (4)$$

where $c$ may generally be a function of time and spatial coordinates. Hereafter, we use a symbol $\partial_\alpha$ for partial derivative with respect to the variables $\alpha = r, z$. Note that, if $v_\theta \equiv 0$, then $\boldsymbol{v} \equiv 0$; that is, a genuinely meridional Beltrami flow does not exist. Note that $rv_\theta$ plays the role of the stream function for the meridional components $v_r$ and $v_z$. Employing the parallelism of velocity and vorticity, the variables $v_r$ and $v_z$ in the second row of (4) can be eliminated to give a kinematic relation between $v_\theta$ and $c$:

$$\partial_r\left(\frac{1}{r}\partial_r(rv_\theta)\right) - \frac{\partial_r c}{c}\frac{1}{r}\partial_r(rv_\theta) + \partial_z^2 v_\theta - \frac{\partial_z c}{c}\partial_z v_\theta + c^2 v_\theta = 0. \tag{5}$$

Furthermore, the incompressible flow must fulfill the continuity condition

$$\nabla \cdot \boldsymbol{v} = \frac{1}{r}\partial_r(rv_r) + \partial_z v_z = 0 \tag{6}$$

Since $\boldsymbol{\omega}$ given by (4) is divergence-free, too, these conditions lead to another one on $c$:

$$\boldsymbol{v} \cdot \nabla c = 0 = \boldsymbol{\omega} \cdot \nabla c \tag{7}$$

$\partial_r c$ and $\partial_z c$ are determined by (5) and (7), respectively, as

$$\begin{aligned}
\partial_r c &= \frac{(\nabla^2 - r^{-2} + c^2)v_\theta}{v_r^2 + v_z^2}v_z \equiv c\Lambda\omega_z, \\
\partial_z c &= -\frac{(\nabla^2 - r^{-2} + c^2)v_\theta}{v_r^2 + v_z^2}v_r \equiv -c\Lambda\omega_r,
\end{aligned} \tag{8}$$

where

$$\Lambda = \frac{(\nabla^2 - r^{-2} + c^2)v_\theta}{\omega_r^2 + \omega_z^2}. \tag{9}$$

Equation (8) implies that, if $c$ is nonzero and constant, then $\Lambda = 0$, provided that $\omega_r \neq 0 \neq \omega_z$. If $c$ depends on either $r$ or $z$ only, either $\omega_r$ or $\omega_z$ may vanish.

## 3. Consistency of Dynamical and Constraint Equations

The two constraints (8) for the partial derivatives of the (pseudo) scalar $c$ are generally incompatible with each other. Consistency of (8) for arbitrary regular solution requires the partial derivatives $\partial_r$ and $\partial_z$ to commute. From (8), a straightforward calculation yields

$$0 = [\partial_r, \partial_z]\ln c = -r\boldsymbol{\omega} \cdot \nabla(\Lambda/r) \tag{10}$$

Let $\phi'$ denote any function that satisfies $\boldsymbol{\omega} \cdot \nabla\phi' = 0$. Then, the function $\Lambda$ given by

$$\Lambda = r\phi', \tag{11}$$

is a solution to (10). In terms of such a function $\phi'$, Equation (9) is expressed as

$$(\nabla^2 - r^{-2} + c^2)v_\theta = r(\omega_r^2 + \omega_z^2)\phi'. \tag{12}$$

On the other hand, employing (11) and referring to (4), the conditions (8) read

$$\partial_r \ln|c| = \phi'\partial_r(rv_\theta), \tag{13}$$

and

$$\partial_z \ln|c| = \phi'\partial_z(rv_\theta). \tag{14}$$

Equations (13) and (14) imply that $c$ will depend on the spatial coordinates via the stream function $\psi \equiv rv_\theta$ only. Notice that $\partial(\phi', \psi)/\partial(r, z) = 0$ for regular $c$ so that $\phi'$ and $\psi$ are mutually dependent when the spatial variations are concerned. We thus may write

$$\ln|c| = \phi(\psi, t) \text{ or } c = c(\psi, t) \tag{15}$$

with a relation $\phi' = \partial\phi(\psi, t)/\partial\psi$. Namely, the conditions (12) with (15) are equivalent to (8). Note that $\psi$ can be time-dependent, too. Recalling the definition (4) of $\omega$, (12) is written as

$$r\partial_r\left(\frac{1}{r}\partial_r\psi\right) + \partial_z^2\psi + c^2\psi = \phi'(\psi, t)\left((\partial_r\psi)^2 + (\partial_z\psi)^2\right) \tag{16}$$

This equation elicits the hidden nonlinearity of the dynamics of the axisymmetric Beltrami vortex when $\phi' \neq 0$ and will henceforth play the central role in our arguments.

Equation (16) governs the spatial variation of the stream function $\psi$ and in this sense corresponds to the Bragg–Hawthorne equation for inviscid flow [47,48]:

$$r\partial_r\left(\frac{1}{r}\partial_r\overline{\psi}\right) + \partial_z^2\overline{\psi} = r^2\frac{dH}{d\overline{\psi}} - \overline{C}\frac{d\overline{C}}{d\overline{\psi}},$$

where $\overline{\psi}$ is the stream function (not necessarily equal to $rv_\theta$), $H \equiv p/\rho + v^2/2$ is called the (pressure or energy) head, and $\overline{C} \equiv rv_\theta \ (= \psi)$ is supposed to be a function of the stream function $\overline{\psi}$ for the axisymmetric inviscid flow. Note that $\overline{\psi}$ and $\overline{C}$ are not related a priori. For recent applications of the Bragg–Hawthorne equation, see, e.g., [17,22,23].

The Bragg–Hawthorne equation is an expression of the equation of motion for inviscid flow under steadiness assumption. Therefore, the solution of this equation is a solution of equation of motion. On the other hand, our Equation (16) has been derived not from the dynamical equations but from the constraints on the unsteady incompressible axisymmetric Beltrami flow. There is no immediate reason that these two equations should coincide with each other except the same derivative terms that result from taking the velocity multiplied by the coordinate as the dependent field. Nevertheless, two equations become essentially identical when they are linear. In fact, assuming that $dH/d\overline{\psi} = 0$ and $\overline{C} \propto \overline{\psi}$, the Bragg–Hawthorne equation writing $d\overline{C}/d\overline{\psi}$ as $c$ (being constant) reduces to (16) with $\phi' = 0$.

Because the advection term is the vector product of vorticity and velocity added by the gradient of the specific kinetic energy, the Navier–Stokes Equation (2) for axisymmetric (i.e., $\partial_\theta = 0$) Beltrami flow is linearized as

$$\partial_t\psi = \nu\left(r\partial_r\left(\frac{1}{r}\partial_r\psi\right) + \partial_z^2\psi\right) + rF_\theta. \tag{17}$$

Equivalently, from (16) and (17), we have

$$\partial_t\psi = \nu\left\{-c^2\psi + \phi'\left((\partial_r\psi)^2 + (\partial_z\psi)^2\right)\right\} + rF_\theta, \tag{18}$$

with $\ln c = \phi$. This is the dynamical equation for $\psi$ that expresses the effect of the dissipation as well as the torque $rF_\theta$ on the rate of temporal change of $\psi$. Noting that $\psi$ is the angular momentum component per unit mass about the symmetry axis, the first term with a negative sign in the large braces of (18) represents the direct resistance on the fluid element. This resistance term, being proportional to $c^2$ and an invariable concomitant of alignment of velocity and vorticity, reveals the hallmark of the Beltrami flow. When $\psi$ is steady and has negligible spatial variations, the direct resistance is balanced with the torque. The second term involves the mixed effects of the rate of strain and vorticity.

The constraint (16) Is written in terms of the velocity component as

$$\nabla^2 v_\theta - \frac{v_\theta}{r^2} + c^2 v_\theta = r\phi'\left\{\left(\frac{1}{r}\partial_r(rv_\theta)\right)^2 + (\partial_z v_\theta)^2\right\}. \tag{19}$$

When $c$ is constant and $\phi' = 0$, it is obvious that the fundamental spatial length scale of the velocity field is given by $c^{-1}$. Many analytic Beltrami flow solutions are known for constant $c$.

It remains to examine in what way the conditions found so far for unsteady axisymmetric Beltrami flows are consistent with the rest of the Navier–Stokes equations, i.e.,

(1) and (3). The details are relegated to Appendix A. One condition is that $1/c(\psi, t)$ is integrable over $\psi$. The components $F_r$ and $F_z$ are related to $c$ through (A2) and (A3) in Appendix A. Then, we have the second condition

$$\nabla^2 H = \nabla \cdot \boldsymbol{F}. \tag{20}$$

When $c$ is constant, (20) can be decomposed to

$$\frac{1}{c}\partial_z F_\theta = \partial_r H - F_r, \tag{21}$$

$$\frac{1}{cr}\partial_r(rF_\theta) = -\partial_z H + F_z. \tag{22}$$

$H$ must be spatially constant if external force is absent. Suppose that $c$ is constant. If either the external force is derived from an axially symmetric potential $V$ or $F_\theta = F_\theta(r) \propto 1/r$, then (21) and (22) immediately lead to Bernoulli's law $H + V$ = constant.

Suppose that $v_\theta$ of a Beltrami flow that obeys the Navier–Stokes Equation (2) is known. Notice that, in solving (2), information of the pressure is not needed because of the system's axisymmetry. It has been shown in Appendix A that the Navier–Stokes Equations (1) and (3) are transformed to (A2) and (A3), and the equations of $\chi \equiv \int^\psi c^{-1}d\psi$. $v_r$ and $v_z$ derived from $\chi$ as described in Appendix A are automatically guaranteed to fulfill the continuity condition as well as the Beltrami relation (4). Equations (A2) and (A3) implicitly involve $p/\rho$ through the head $H$. Therefore, what we need to obtain $v_r$ and $v_z$ from $\chi$, besides the boundary conditions which are compatible with the Beltrami relation (4), are $H$ and the external force $\boldsymbol{F}$ related by (20) or (21) and (22). Once each component of $v$ is determined as a function of $r$, $z$, and $t$, $c$ is obtained from the Beltrami relation. That this procedure is possible is a peculiarity of the axisymmetric Beltrami flow. $\boldsymbol{F}$ will be present when $c$ is not constant. The details on the relation of nonconstant $c$ and $\boldsymbol{F}$ are elaborated in Appendix B.

## 4. Vortex Solutions

In this section, we consider viscous Beltrami vortices with constant $c$ (Trkalian flow) and nonconstant $c$ separately. The Beltrami flows, inviscid or viscous, with constant $c$ were studied in many works, as was cited in Introduction. The case in which $c$ is variable is studied formally in [46] for axially asymmetric flows in a Cartesian coordinate system. In particular, if $c$ has no explicit time dependence and the external force is conservative and the Bernoulli function is constant, then $v^2$ decays timewise exponentially. The generalized ABC flow solution for compressible flow was also found for $c$ that is dependent on a single spatial variable in [34].

### 4.1. Constant c

We first solve the dynamical equations with the constraints of constant $c$ and $F_\theta = 0$. From (13) and (14), $c$ can be constant when $\psi$ is constant or $v_\theta \propto 1/r$, too. However, this does not satisfy (16).

In case $\psi$ is not constant, constant $c$ implies $\phi' = 0$. Then, (19) is linear and homogeneous. It is easily solved for a single mode flow by the separation of variables:

$$v_\theta = \frac{\psi}{r} = \alpha(t)J_1(\Lambda r)\sin(kz + \delta), \quad \Lambda = \sqrt{c^2 - k^2}, \tag{23}$$

where $J_1$ is the Bessel function of the first order, $\alpha(t)$ a function of $t$ to be determined later and $\delta$ a phase. $k$ and $\delta$ are both real or pure imaginary. If we require the solution to be nondivergent in the whole spatial region, $k$ must be real and $|k| < c$. Single mode solutions of this type frequently appear, irrespective of the symmetry when the nonlinear terms are somehow removed [22,35]. See also [49], wherein the Coriolis force is taken into account. From (23) and (4), it follows that

$$v_r = -\frac{k}{c}\alpha(t)J_1(\Lambda r)\cos(kz+\delta), \quad v_z = \frac{\Lambda}{c}\alpha(t)J_0(\Lambda r)\sin(kz+\delta). \tag{24}$$

Referring to (4), it is readily confirmed that the equality $\omega = cv$ holds.

Equation (18) gives

$$(\partial_t + \nu c^2)\alpha(t) = 0. \tag{25}$$

Since $c \neq 0$, this equation has a nontrivial solution

$$\alpha(t) \propto e^{-\nu c^2 t}.$$

The time-dependent velocity fields given by (23) and (24) then fulfill the Navier–Stokes Equations (1)~(3). As was expected, solutions exist whose kinetic energy density $v^2/2$ is finite in the whole space.

The characteristic decay time is given by $1/\nu c^2$. Only steady vortex solutions are allowed in the inviscid limit. Separation of the variables $r$ and $t$ in the above solution is a consequence of breaking the scaling invariance of the Navier–Stokes equations. Exact three-dimensional solutions with the decay-time inversely proportional to the kinematic viscosity have been presented by several authors [11,35,36,50].

*4.2. Nonconstant c*

The proportionality coefficient $c$ will be variable when $F$ is nonconservative or $H + V$ is not spatially constant. Unfortunately, we do not know the functional form of $c(\psi, t)$ or $\phi(\psi, t)$, whose temporal evolutions will be governed by the dynamics. However, it may be sensible to set their instantaneous form as an initial condition. We consider a simplest case given by

$$\phi(\psi, t) = \ln A + B\psi, \text{ i.e., } c(\psi, t) = Ae^{B\psi}, \tag{26}$$

where $A$ and $B$, whose dimensions are [length$^{-1}$] and [length$^{-1}$·velocity$^{-1}$], respectively, are spatially constant (but possibly time-dependent). The assumption (26) amounts to keeping the first two terms in the power series expansion of $\phi$ in $\psi$. With this choice, the scales of the spatial extension and velocity of the solution will be given by $A^{-1}$ and $A/B$, respectively. The turn-over time of the vortex rotation will be the order of $B/A^2$. The representative Reynolds number is given by Re $= 1/\nu B$.

In passing, the assumption of time-independent $c$ employed in [46] and the form (26) are compatible with each other for a single mode solution whose stream function decays as $e^{-\nu c^2 t}$. In this case, the parameter $B$ may be chosen as to be proportional to $e^{\nu c^2 t}$ to cancel the decay factor $e^{-\nu c^2 t}$. Investigating whether this is possible or not for axisymmetric flow is beyond the scope of the present paper.

We begin with exploring the spatial dependence of the velocity fields with the time fixed. Since $\phi' = B$, we rewrite Equation (16) as

$$\left(\partial_r^2 - \frac{1}{r}\partial_r\right)\psi + \partial_z^2\psi + A^2 e^{2B\psi}\psi = B\left\{(\partial_r\psi)^2 + (\partial_z\psi)^2\right\}. \tag{27}$$

Changing the variable by $B\psi = -\ln|g|$, Equation (27) is put into the form

$$\left(\partial_r^2 - \frac{1}{r}\partial_r\right)g + \partial_z^2 g + A^2\frac{\ln|g|}{g} = 0. \tag{28}$$

The analytic solution of (28) is not known. Since the term $A^2 g^{-1}\ln|g|$ results from a variation of $U(g) = A^2\left(\ln|g|\right)^2/2$, we may approximate it around the minimum point $g = 1$ by a harmonic potential $U(g) \approx A^2(g-1)^2/2$ and linearize (28) as

$$\left(\partial_r^2 - \frac{1}{r}\partial_r\right)\hat{g} + \partial_z^2\hat{g} + A^2\hat{g} = 0, \tag{29}$$

where $\hat{g} \equiv 1 - g \approx B\psi$. Equation (29) is identical to (19) with $\phi' = 0$. Therefore, when $|\hat{g}|$ is sufficiently smaller than unity, the solution to (28) is given by (23), with $c$ replaced by $A$. If $v_\theta$ decreases as fast or faster than $1/r$ for $r \to \infty$, the flow is approximately given by (23) and (24). The velocity decays exponentially with time.

Equation (27) permits a variety of solutions. Shown in Figure 1 are three examples, (i), (ii), and (iii), obtained numerically under the condition that $|\partial_z \psi| = |\partial_z^2 \psi| = 0$. That is, these solutions have no $z$-dependence. Each solution for $v_\theta$ has one peak and decays gradually at long distances. Equations (28) or (29) show that $1/|A|$ defines the system's length scale, so that, grossly speaking, the (half) width of the peak becomes larger with the decrease in $A$. From Figure 1, the radial distance to the peak and the azimuthal velocity at the peak are grossly read off as $0.5 A^{-1}$ and $A|B|^{-1}$, respectively. Taking these as the rules of thumb, the turn-over time of the vortex at the peak is given grossly by $2\pi(0.5A^{-1})/(A|B|_{-1}) \sim O(|B|/A^2)$, as was already expected. The existence of the solutions (i), (ii), and (iii) implies the existence of an infinite number of continuous sequences of solutions in accordance with the continuous change in the parameters $A$, $B$, and the boundary condition $\partial_r \psi(r_0)$.

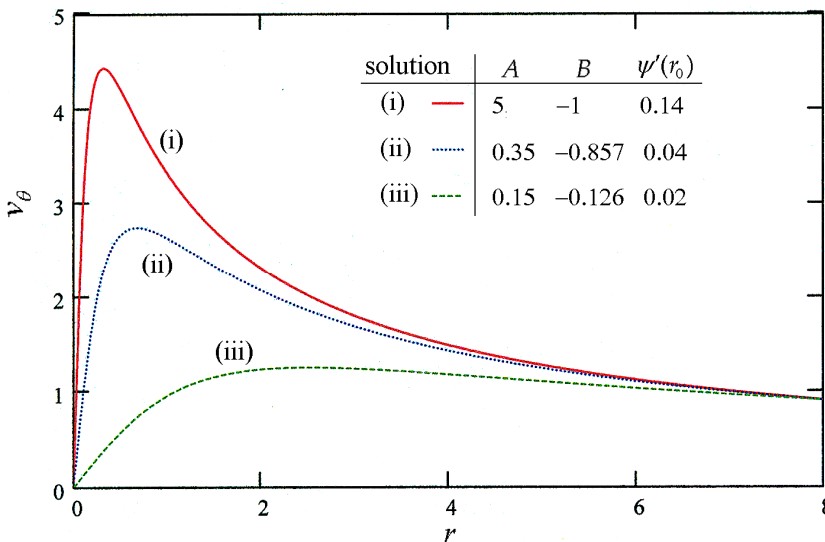

**Figure 1.** Numerical solutions of (27) under the conditions $\partial_z \psi = \partial_z^2 \psi = 0$. Ordinate is $v_\theta = \psi/r$. Three curves, (i), (ii), and (iii), are the solutions for three sets of the parameters $A$, $B$, and $\partial_r \psi(r_0)$, as designated in the figure. Each $\psi(r_0)$ is chosen so that the straight extrapolation of the curve directs the origin. Integrations were performed in a region $r_0 = 0.002 \le r \le 8$.

It is easily confirmed by substitution that, ignoring the last term on the l.h.s. of (27), the solution to (27) for large $r$ and fixed $z$ is approximated by $\psi \approx \psi_0 \equiv -(2/B) \ln r$ with negative $B$. The tail of the solution is therefore approximated by

$$v_\theta \approx -(2/B) \ln r/r,$$

which, from (26), implies that $c$ is nearly proportional to $1/r^2$ at long distances. Vortex solutions with such a long tail were numerically confirmed to exist up to $r = 30$ for $|A| \le 6.84\cdots$. We disregard the case of larger $|A|$, for which, although not shown here, numerical calculations exhibited that the solutions keenly oscillate around $z = 0$.

As regards the behavior in the finite $z$ region, Equation (27) suggests that nondivergent $\psi$ will be expanded in a power series of $e^{-|A|z}$. By substituting the expression

$$B\psi(r, z, t) = \sum_{n=0}^{\infty} \psi_n(r, t)\, e^{-n|A|z}, \quad \psi_0 \ne 0, \tag{30}$$

to (27) and comparing the coefficients of $e^{-n|A|z}$, the equations of $\psi_n$ are, to $n = 2$,

$$
\begin{aligned}
&\psi_0'' - \tfrac{\psi_0'}{r} + A^2 c_0^2 \psi_0 = \psi_0'^2, \\
&\psi_1'' - \tfrac{\psi_1'}{r} + A^2 \big(1 + c_0^2(1 + 2\psi_0)\big)\psi_1 = 2\psi_0'\psi_1', \\
&\psi_2'' - \tfrac{\psi_2'}{r} + A^2\big\{\big(4 + c_0^2(1 + 2\psi_0)\big)\psi_2 + 2c_0^2(1 + \psi_0)\psi_1^2\big\} = 2\psi'_0\psi'_2 + \psi'_1{}^2 + A^2\psi_1^2,
\end{aligned}
\tag{31}
$$

where $c_0 \equiv e^{\psi_0}$ for the choice (26) for $c(\psi)$. The prime stands for a differentiation with respect to $r$. The first equation in (31) gives the solutions shown in Figure 1. Specifically, $\psi_0 \sim -2\ln r$ for large $r$.

Splitting the term $A^2 c_0^2 \psi_n$ as the sum of linear and nonlinear terms by $A^2 \psi_n + A^2(c_0^2 - 1)\psi_n$ and discarding all the nonlinear terms, the equations in (31) for $n \geq 1$ may be linearized as

$$
\psi_n'' - \frac{\psi_n'}{r} + A^2(n^2 + 1)\psi_n = 0.
$$

The $n^2$ emerges from $\partial_z^2$ in (27). This means that $\psi_n/r$ obeys the Bessel differential equation. Thus, restricting ourselves to regular solutions to (31), $\psi_n$ will behave as $rJ_1\big((n^2 + 1)^{1/2}\big|A\big|r\big)$ both near $r = 0$ and at medium distances provided that the non-linear terms are negligible.

At very large distances, the term involving the derivative of $\psi_0$ on the r.h.s. will have to be retained. Thus, for the leading form of the solutions, we solve the equation

$$
\psi_n'' + \frac{3}{r}\psi_n' + 2A^2\psi_n = 0.
$$

Regular solutions exist and are given by $r^{-1}J_1\big((n^2 + 1)^{1/2}|A|r\big)$.

The expected exponential behavior in $z$ specified by (30) can be observed analytically in the vicinity of the symmetry axis, as is shown below.

Near the $z$-axis, $v_\theta$ behaves as $v_\theta \propto r$. With the time being fixed, let us put

$$
\psi = a(z)r^2 + O(r^4),
\tag{32}
$$

for $r \approx 0$. Substituting (32) to (27), we have

$$
a'' + A^2 a + (\text{contribution from } O(r^4) - \text{term in } \psi) = 4Ba^2,
\tag{33}
$$

where prime stands for a derivative with respect to $z$. Referring to Figure 1, it will not be unreasonable as a first approximation to ignore the $O(r^4)$-term at the very vicinity of $z$-axis. Multiplying $\partial_z a$ to the both sides of (33) and integrating once, we have

$$
a' = \pm\sqrt{C - A^2 a^2 + (8B/3)a^3},
\tag{34}
$$

where $C$ is an integration constant, which is generally time-dependent. Solutions to (34) may be expressed in terms of Jacobi's elliptic function or are easily found via numerical integration. Here, we give an exact solution that is constructed by elementary functions for a special combination of the parameters, i.e., $48B^2C = A^6$. In this case, the polynomial in the square root in (34) is factorized to $(8B/3)(a + q)^2(q/2 - a)$, $q = A^2/4B$. Using the integration formula

$$
\int \frac{dx}{(x + \alpha)\sqrt{\beta - x}} = \frac{2}{\sqrt{\alpha + \beta}}\ln\frac{\sqrt{\alpha + \beta} - \sqrt{\beta - x}}{\sqrt{x + \alpha}},
$$

with $\alpha = q = A^2/4B$ and $\beta = q/2 = A^2/8B$, we have

$$a(z) = \frac{A^2/4B}{1+G(z)}\left[\sqrt{\left(2G(z) + \frac{3}{1+G(z)}\right)\left(4 + \frac{3}{1+G(z)}\right) + 3} - 2 - G(z) - \frac{3}{1+G(z)}\right] \tag{35}$$

where

$$G(z) \equiv (2 + \sqrt{3})e^{Az}. \tag{36}$$

The numerical factor in front of $e^{Az}$ in (36) has been chosen for a convention $a(0) = 0$. (Recall that the origin of the coordinate $z$ is arbitrary.) As was already expected, $a(z)$ has a power series expansion in $e^{-|A|z}$ and converges exponentially to a constant value as $z \to \infty$ near the symmetry axis.

The quantity $4a(z)/A^2$ as a function of $z$ was numerically calculated, and the result is shown in Figure 2 for four cases: $A = -10, -5, 10$, and $5$. When $A$ is positive, $|a|$ monotonically increases with $z$, while a negative $A$ gives rise to an additional zero, $z_0$. Numerical calculation gives $|A|z_0 = 2.6391\cdots$, which is independent of $|A|$ because $a(z)$ is a function of $Az$ only. At this point $v_z$ vanishes. In each case depicted in the figure, $4a/A^2$ exponentially approaches a constant, either 1 (exact value) or $0.101020\cdots$, for $z \to \infty$.

Note also that, when $A$ is negative, there is a $z$ at which $da/dz = 0$ or $v_r = 0$ from (4). The distance of the stream line from the symmetry axis minimizes at this $z$ and then increases to infinity as the stream line approaches the plane $z = 0$ or $z_0$. The case of negative $A$ may be suited to describing a flow between two parallel planar boundaries. As regards the solutions depicted in Figure 2, the flow directing toward the symmetry axis just below the plane $z = z_0$ forms a downdraft (i.e., $a(z) < 0$) near the axis and then changes the direction to far infinity near $z = 0$. The flow above the plane $z = z_0$ forms an updraft ($a(z) > 0$). These characteristics of the flows will be altered if the solutions are extrapolated to negative $z$.

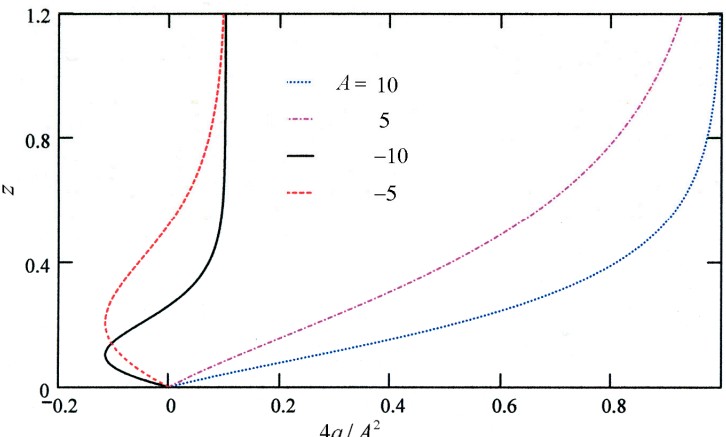

**Figure 2.** The (ordinate)-dependence of $4a/A^2$ (abscissa), where $a(z, t) \approx \psi/r^2$ near the symmetry axis. Boundary condition is $a(0, t) = 0$.

### 4.3. Time Dependence

All of the calculations of the velocity field so far are at some fixed time. In order to determine the time-dependence of the solutions, we start from the fact that the time-dependent solution to the linear equation of motion (17) is generally expressed as a superposition of the single-mode solutions as

$$\psi = r\int \widetilde{v}(q, k, t)J_1(qr)e^{ikz}qdqdk + r\int G_0(\sigma; \sigma')F_\theta(\sigma')d\sigma', \tag{37}$$
$$\widetilde{v}(q, k, t) = f(q, k)e^{-\nu(q^2+k^2)(t+t_0)},$$

where $\sigma \equiv (r, z, t)$ and $\widetilde{v}(q, k, t)$ is the Fourier–Hankel transform of $v_\theta = \psi/r$ with the integration volume element $d\sigma = rdrdzdt$. The integration regions are such that $0 \leq q < \infty$, $-\infty < k < \infty$, $0 \leq r < \infty$, $-\infty < z < \infty$, and $-\infty < t < \infty$. The retarded Green's function $G_0(\sigma; \sigma')$ satisfies, together with (A6),

$$\left\{\partial_t - \nu\left(\partial_r^2 - \frac{1}{r}\partial_r + \partial_z^2\right)\right\} rr' G_0(\sigma; \sigma') = \delta(t - t')\delta(r - r')\delta(z - z'),$$

and is explicitly given in Appendix B. Note that $\tilde{v}(q, k, t)$ has a Gaussian damping factor with respect to $q$ and $k$. At the same time, it exhibits a temporally exponential decay. Spectrum function $f(q, k)$ is the remaining factor of $\tilde{v}$.

The $\tilde{v}(q, k, 0)$ will be determined from the Fourier–Hankel transformation of $v_\theta$ obtained (numerically) by solving (27) or (28) for an unknown $t_0$ and by extracting the Gaussian damping factor from the solution at large $q$ or $k$. Finally, the time dependence is recovered by substituting the damping factor $\exp(-\nu(q^2 + k^2)(t + t_0))$ as in (37).

Two hypothetical cases will be instructive. If $f(q, k)$ distributed on a "circle" $q^2 + k^2 = c^2$, (37) would be a superposition of (25), allowing the possibility of pure imaginary $q$ and/or $k$. If $f(q, k)$ depended on $q$ as $q^{\mu-1}$, the integration over $q$ in (37) would be performed to give the $r$-dependence

$$v_\theta \propto (r/(\nu t)^{\mu/2+1})_1 F_1(\mu/2 + 1; 2; -r^2/4\nu t),$$

where $_1F_1$ is the confluent hypergeometric function, whose limiting behavior is given by

$$_1F_1\left(\frac{\mu}{2} + 1; 2; -\frac{r^2}{4\nu t}\right) \rightarrow \begin{cases} 1 - (\mu + 2)r^2/16\nu t, & r \rightarrow 0, \\ (2 - \mu)(r^2/4\nu t)^{-\mu/2-1}/2, & r \rightarrow \infty. \end{cases}$$

Note that, in the above example, the combination of $r$ and $t$ as the similarity variable $r^2/\nu t$ emerges in the velocity field. This phenomenon results from some distribution of the wavenumber and indicates the partial restoration of the scaling invariance that had once been broken by setting $c$ as constant.

The procedure of extracting the Gaussian factor and the spectrum function $f$ from the data for $v_\theta$ of the solution (i) in Figure 1 is sketched in Appendix C, and the result is shown in Figure 3. In the present case, the $q$-dependence of $\tilde{v}$ depicted in Figure 3 indicates $f(q) \rightarrow 1$ for $20 < q^2 < 40$. The spectrum function of the solution (i) is prominent at small $q$ or long wavelengths and exhibits a Gaussian damping at short wavelengths.

The spectrum function shown in Figure 3 is employed to recover the time-dependence of the solution (i) in accordance with (37), with the result shown in Figure 4. In the numerical calculations, the upper bound of each integration was taken to give the tenth zero of $J_1$ in the integrand. Therefore, calculational errors accumulated near $r = 0$. It is notable that the resultant solution temporally behaves similarly to the familiar scale invariant solutions [4,5,9]. See also Figure 5.17 in [10] for the (two dimensional) scaling invariant Oseen's flow.

Unfortunately, the $\psi$ constructed in this way does generally not satisfy the constraint Equation (16) for $t > 0$ because the constraint equation is entirely independent of the equation of motion. Thus, some temporal changes in the tilt of vorticity from the one in the Beltrami vortex will inhere in the $\psi$ given by (37).

However, we can always choose a continuous and differentiable sequence of solutions to the constraint equation. This is because the differential Equation (27) is not singular. Let the sequence start from, e.g., the solution (i) in Figure 1. We specify each solution on the sequence with a labelling parameter $t_0$ and the spectrum function $f(q, k; t_0)$. Shifting the parameter as $t_0 \rightarrow t + t_0$ with $t_0$ to be used for the solution (i), the solution $t$ to the constraint equation is written as

$$\psi_C(r, z; t) = r\int_0^\infty f_C(q, k; t + t_0)e^{-\nu(q^2+k^2)(t+t_0)} J_1(qr)e^{ikz} q\,dq\,dk.$$

The suffix C means that it is the solution to the constraint equation. The $t$-dependence of $f_c$ reflects the $t$-dependences of the parameters $A$, $B$, and $\partial_r\psi(r_0)$, too.

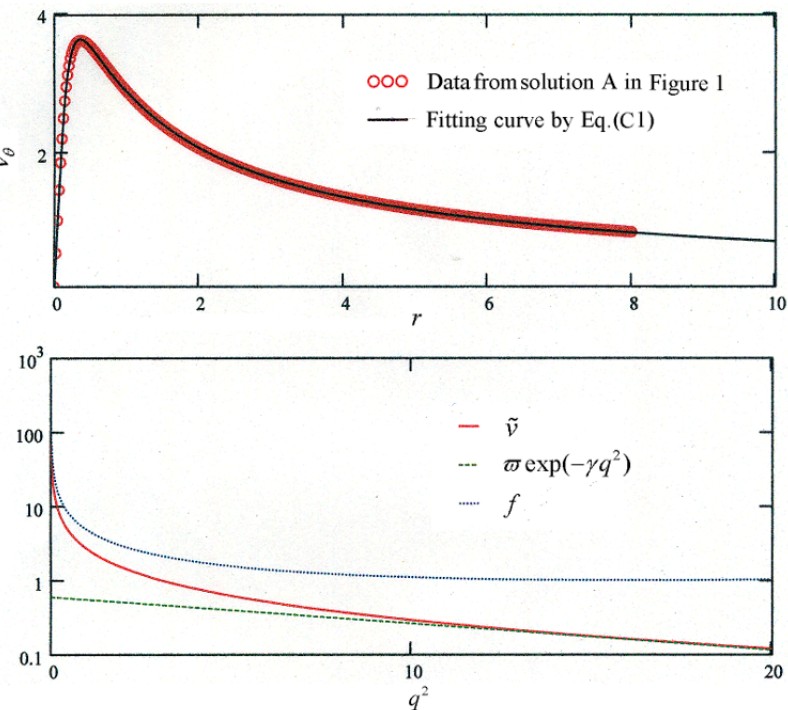

**Figure 3.** Upper panel: The numerical solution (i) in Figure 1 is shown by red circles and red thick curve. Solid black curve is the result of fitting to the numerical solution (i) by Equation (A9) in Appendix C. Lower panel: Hankel transform $\tilde{v}$ (red solid curve), Gaussian factor $\varpi exp(-\gamma q^2)$ with $\varpi = 0.28$ and $\gamma = 0.048$ (green broken line) and $f = \tilde{v}(q, 0)exp(\gamma q^2)$ (blue dotted curve).

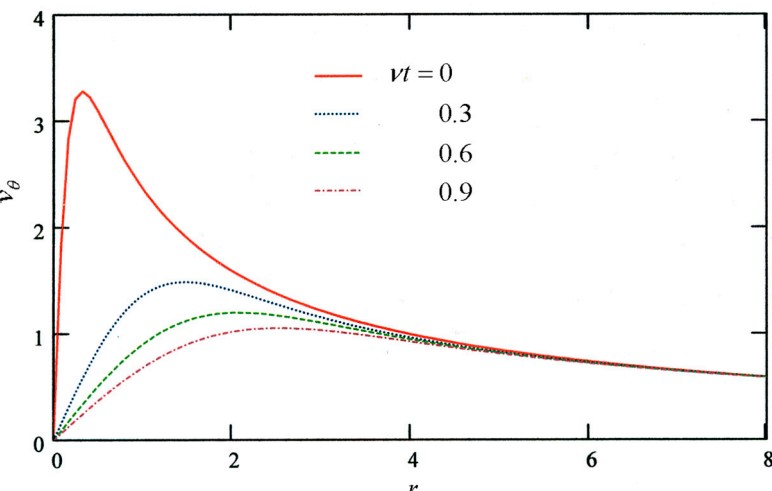

**Figure 4.** Temporal variation due to (15) with $F_\theta = 0$ of the solution (i) for $v_\theta$ in Figure 1. Integration was performed up to the tenth zero of $J_1$.

On the other hand, the solution to the evolution equation with the initial condition chosen as, e.g., solution (i) in Figure 1, is given by (37):

$$\psi_D(r, z, t) = r\int_0^\infty f_D(q, k)e^{-\nu(q^2+k^2)(t+t_0)}J_1(qr)e^{ikz}qdqdk + r\int G_0(\sigma; \sigma')F_\theta(\sigma')d\sigma',$$

where the suffix D stands for being the solution to the dynamical equation. The above expression for $\psi_D$ is obtained by the Green function method for the operator $\partial_t - \nu(\partial_r^2 - r^{-1}\partial_r + \partial_z^2)$ similar to the one described in Appendix B.

Now, identifying the labelling parameter $t$ in $\psi_C(r, z; t)$ with the physical time, we ask a question: Under what condition does the $\psi_C(r, z; t)$ coincide with the $\psi_D(r, z; t)$?

This question seems reasonable because of the similarity of Figure 1 to Figure 4. If $\psi_C(r, z; t) = \psi_D(r, z; t)$, operating $\partial_t - \nu\nabla^2$ to $(\psi_C(r, z; t) - \psi_D(r, z; t))/r$ from the left and using the identity (B3), we have

$$F_\theta(r, z, t) = \int_0^\infty \partial_t f_C(q, k; t + t_0)e^{-\nu(q^2+k^2)(t+t_0)}J_1(qr)e^{ikz}qdqdk. \qquad (38)$$

This is the (necessary) condition for the Beltrami vortex to continuously exist. The procedure of constructing $\psi_C$ described above implies that $f_C(q, k; t + t_0)$ is $t$-dependent and the r.h.s. of (38) does not generally vanish, so that $F_\theta$ does not vanish either. The analysis that has led to this consequence is based on the assumption (26) for $c(\psi, t)$. Whether a sequence whose $f_c$ does not depend on $t$ exists for a more general form of $c(\psi, t)$ is an open question.

In summary, an axially symmetric incompressible viscous swirling Beltrami vortex with variable $c$ is possible when an external force is present.

## 5. Conclusions and Discussion

In the axisymmetric viscous swirling Beltrami vortices, the proportionality coefficient $c$ between the velocity and vorticity is constrained by the equation analogous to the Bragg–Hawthorne equation for steady inviscid flow. Our consistency equation, applicable also to unsteady flows, is nonlinear in the stream function except for the case of a constant $c$. The exact solutions found for a constant $c$ are of the type known up to now for inviscid flow [22,49].

One of the main results for the case of a nonconstant $c$ is that the spatial and temporal dependence of $c$ emerge through the stream function $\psi = rv_\theta$, i.e., $c = c(\psi, t)$. The physically meaningful numerical solution sought in the simplified form of $c(\psi, t)$ exhibited a cyclone-like behavior for radial motion and a gradual temporal decay. The velocity field does not diverge at infinite distances, as has been shown analytically or numerically in the present study, in which $\ln|c|$ is taken to be linear in $\psi$.

Breaking the scale invariance due to $c$ of the dimension [length$^{-1}$] generates a characteristic length $l = c^{-1}$ and a decay time $\tau = l^2/\nu$ for a single-mode flow. It should be noted that the decay time does not depend on the wavenumber, $k$, appearing in (23). For air at normal pressure and room temperature, $\tau_{air} \approx 5 \times 10^4 (l/m)^2$s. Similarly, for water, we have an estimation $\tau_{water} \approx 10^6 (l/m)^2$s. These periods shall be long enough for a macroscopic Beltrami vortex to persist. We also saw that a nonconstant $c$ will result in a longevity of the vortex when the spectrum function obeys a power law.

The second important finding is that an axisymmetric Beltrami vortex with a nonconstant $c$ continues to be Beltramian only when some special kind of external force depending on the consistency equation must be at work. For example, the $F_\theta$ acts to directly grow the azimuthal component of the flow and induce the meridional components of the motion via the stream function. This is why the vortices undergoing such external forces as the Coriolis force or the Lorentz force are worth studying. The vortex–boundary and vortex–vortex interactions will also play the role. The numerical simulation method is expected to provide powerful tools to elicit information on this issue [27,42–44].

Without the first principle to determine the functional form of $c(\psi, t)$ or **C**, in the present paper, we have examined only the simplest form of the consistency condition for the viscous vortices. Numerous types of consistency conditions remain to be explored.

**Funding:** This research received no external funding.

**Data Availability Statement:** The data that support the findings of this study are available from the corresponding author upon reasonable request.

**Acknowledgments:** The author thank S. Kono at Tohoku University for his courteous support to the present study.

**Conflicts of Interest:** The author has no conflict to disclose.

## Appendix A. Consistency of Constraints with the Navier–Stokes Equations (1) and (3)

Since $c$ depends on $r$ and $z$ through $\psi$ only, one can write

$$\partial_\alpha \psi / c = \partial_\alpha \chi, \ \alpha = r \text{ or } z \tag{A1}$$

where $\chi \equiv \int_0^\psi c^{-1} d\psi$. Since $c = c(\psi(r, z, t), t)$ is a function of $\psi$ and $t$, so is $\chi$. Due to the Beltrami condition (4), this implies $v_r = -\partial_z \chi / r$, $v_z = \partial_r \chi / r$ and $\nabla \cdot v = 0$. $c$ is constant if $\chi$ is proportional to $\psi$. Since the advection term in the Navier–Stokes equations is written as $(\boldsymbol{\omega} \times v)_\alpha + \partial_\alpha v^2 / 2$, the Navier–Stokes Equations (1) and (3) with the relation (4) are rewritten for the Beltrami flow as

$$\partial_t \frac{\partial_z \chi}{r} = \boldsymbol{\nu} \left( \nabla^2 - \frac{1}{r^2} \right) \frac{\partial_z \chi}{r} + \partial_r H - F_r, \tag{A2}$$

$$\partial_t \frac{\partial_r \chi}{r} = \boldsymbol{\nu} r \nabla^2 \frac{\partial_r \chi}{r} - \partial_z H + F_z, \tag{A3}$$

respectively. Here, $H$ is the head. Equations (A2) and (A3) are employed to determine $F_r - \partial_r H$ and $F_z - \partial_z H$, respectively, when $\chi$ is known. This is possible only when these equations are compatible with each other as the differential equations of $\chi$. To examine the compatibility, noting that $\nabla^2 = r^{-1} \partial_r (r \partial_r) + \partial_z^2$ and $\nabla^2 - r^{-2} = \partial_r (r^{-1} \partial_r r) + \partial_z^2$, operate $\partial_r$ to (A2) and $\partial_z$ to (A3) from the left. Subtracting each side of the resultant equations, one obtains the consistency condition (20) in the text.

Suppose that $c$ is constant so that $\chi = \psi / c$. Divide both sides of (17) by $c$ and subsequently operate $\partial_z$ from the left. The resultant equation and (A2) are employed to eliminate the time-derivative term to obtain (21) in the text. Similarly, operating $\partial_r$ from the left of (17) and employing (A3) to eliminate the time-derivative term, we have (22) in the text.

## Appendix B. Variable $c$ Generally Implies Time-Dependent External Force

When the condition (18) is fulfilled, (A2) and (A3) are equivalent so that $\chi$ is obtained by solving either (A2) for $\partial_z \chi / r$ or (A3) for $\partial_r \chi / r$. Here, we take (A3). Then $\partial_r \chi / r$ is expressed as a sum of homogeneous and inhomogeneous terms as

$$\frac{\partial_r \chi}{r} = \text{Re} \int \pi_0(q, k) e^{-\nu(q^2 + k^2)t} J_0(qr) e^{ikz} q dq dk + \int G_0(\sigma; \sigma') S_r(\sigma') d\sigma', \tag{A4}$$

where $\sigma = (r, z, t)$ and $\int d\sigma' = \int_0^\infty r' dr' \int_{-\infty}^\infty dz' \int_{-\infty}^\infty dt'$. $J_0$ is the Bessel function, and $\pi_0(q, k)$ is an arbitrary function of $q$ and $k$ to be determined by boundary condition. $S_r(\sigma)$ is defined by

$$S_r(\sigma) \equiv \partial_r H(\sigma) - F_r(\sigma), \tag{A5}$$

$G_0(\sigma; \sigma')$ is the retarded Green's function that satisfies

$$(\partial_t - \boldsymbol{\nu} \nabla^2) G_0(\sigma; \sigma') = \frac{1}{r} \delta(r - r') \delta(z - z') \delta(t - t'), \tag{A6}$$

and is given for the present problem by

$$G_0(\sigma; \sigma') = \frac{\theta(t - t')}{4\sqrt{\pi \boldsymbol{\nu}^3 (t - t')^3}} \exp\left( -\frac{r^2 + r'^2 + (z - z')^2}{4\nu(t - t')} \right) I_0\left( \frac{rr'}{2\nu(t - t')} \right), \tag{A7}$$

Here, $\theta(t)$ is the Heaviside step function and $I_0$ the modified Bessel function of the first kind. Equations (A6) and (A7) are verified by using the formulae

$$\int_0^\infty J_1(qr) J_1(qr') q dq = \frac{1}{r} \delta(r - r'),$$
$$\int_0^\infty J_1(qr) J_1(qr') e^{-\tau q^2} q dq = \frac{1}{2\tau} e^{-(r^2 + r'^2)/4\tau} I_0\left( \frac{rr'}{2\tau} \right).$$

The Fourier–Hankel transform of $G_0$ given by

$$\int G_0(\sigma;\, \sigma') \mathrm{e}^{ikz'} J_0(qr') r' dr' dz' = \frac{1}{2}\theta(t-t') \mathrm{e}^{-\nu(q^2+k^2)(t-t')+ikz} J_0(qr)$$

can be used to obtain an explicit relation among $\chi$, $\pi_0$, and $S_r$. Equating $\partial_r \chi / r$ given by (A4) with $\partial_r \psi / rc$ and operating $\partial_t - \nu \nabla^2$ from the left, we have

$$S_r(\sigma) = \left(\partial_t - \nu \nabla^2\right) \frac{\partial_r \psi}{cr}, \tag{A8}$$

Therefore, in case $\psi$ and $c$ are not spatially constant, $S_r(\sigma)$ given by (A5), in particular $F_r$ when $H$ is spatially constant, is required to be nonzero. Since $\psi$ depends on time, so does $F_r$. In this case, $F_z$ is determined from the constraint (18).

### Appendix C. Extracting the Gaussian Factor and the Spectrum Function from the Solution (i) in Figure 1

As an illustration of the calculational procedure, we assume that there is no $z$-dependence. This amounts to adopting the zeroth order term in the expansion (30) in terms of $e^{-|A|z}$. Considering that $v_\theta$ is linear in $r$ near $r = 0$ and behaves as $\ln r / r$ at large distances, we fit the solution (i) in Figure 1 by

$$v_{\theta,\,\mathrm{fit}}(r) = b_0 \frac{1 - \mathrm{e}^{-b_1 r^2}}{r} + b_2 \frac{\ln(1 + b_3 r^2)}{r}. \tag{A9}$$

The first term on the r.h.s. is borrowed from [1] for the familiar exact solution. Choosing the parameters as $b_0 = -0.2563$, $b_1 = 3.736$, $b_2 = 0.9403$, $b_3 = 26.93$, (A9) reproduces the data quite well, as is shown in the upper panel of Figure 3 in the text.

The Hankel transform $\widetilde{v}$ of $v_\theta$ approximated by the analytic expression (A9) is numerically calculated and is shown by a red solid curve in the lower panel of Figure 3. The Gaussian factor $\exp(-\gamma q^2)$ is read off from large $q$ behavior of $\widetilde{v}$. The spectrum function $f(q)$ at a fixed time is obtained by multiplying $\exp(\gamma q^2)$ to $\widetilde{v}$. Time-dependent $v_\theta$ is constructed through the inverse Hankel transform of $f(q) \exp(-\nu q^2 t)$ with the result shown in Figure 4. If $z$-dependence is also known, the corresponding Fourier transform is also employed to obtain the full time dependence.

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
