# Peer review of "Three-Dimensional Unsteady Axisymmetric Viscous Beltrami Vortex Solutions to the Navier–Stokes Equations"

_2571-8800, doi:10.3390/j6030030_

Round 1

Reviewer 1 Report

Please see the attached PDF file.

Following the instructions I found on the journal's webpage, I did not assess the quality of the manuscript. However I detected places where minor improvements are needed and will leave it to the editors to deal with those. Overall the material is well explained and should be easy to follow from the language point of view.

Author Response

Author’s reply to Reviewer_1’s comments

  1. Takahashi

I appreciate many valuable comments given by the reviewer 1. My replies are given in the followings for every item of the comments. In the revised manuscript, the corrected or modified parts are designated by red fonts. The equation numbers and line numbers below are those appearing in the revised manuscript.

 Unfortunately, I could not obtain the suggested reference “An exact anelastic Beltrami-fl ow solution for use in model validation…”, so that I did not include it to References.

  1. line 150: Correction:: “incompressible fluid” à “incompressible flow”

2., 4. lines 161-168: I intended that the symbols  or  collectively express the solution of . I modified the relevant sentences as below:

Then the function  given by

,

(11)

is a solution to (10). In terms of such a function , equation (9) is expressed as ….

The exposition leading to (15), which is based on the fact , was modified. The more detailed argument, which I omitted in the paper, goes as follows. Suppose that  and  are regular functions of  and . Differentiate (13) and (14) with respect to  and , respectively, and subtract the both sides of the resultant equations from each other. We have the vanishing Jacobian

,                             (R1)

which gives

                     (R2)

for nonvanishing . The last equality is due to (R1) and .

Suppose that  depends on the spatial coordinates through mutually independent functions  and , i.e., the nonvanishing variation  is not trivially related to . The change in spatial coordinate  brings about the following variation of :

                            (R3)

Comparing (R3) with (R2), we have . That is,  does not have a dependence on function other than . Variation in  leads to the same consequence. The freedom of adding an arbitrary function of  only is absorbed to a freedom of redefinition of .

  1. See the revised manuscript for the new numberings of the equations (13) and (14).
  2. lines 173-189: The relation between equation (16) and the Bragg-Hawthorne equation is explained below (16).
  3. line 200: I adopted the expression “When is steady and has negligible spatial variations,”
  4. line 201: The expressions (12) and (16) are equivalent. I followed the reviewer’s suggestion.
  5. lines 213-214: I followed the reviewer’s suggestion.
  6. lines 216-227: It has been shown in Appendix A that the Navier-Stokes equations (1) and (3) are transformed to the equations of , (A2) and (A3), which involve through the head . Therefore, what we need to obtain and  from  are, besides the boundary conditions,  and the external force  related by (20) or (21) and (22). This is the peculiarity of the axisymmetric Beltrami flow.

Although there is no circular logic, I admit that the exposition in the last paragraph of Section 3 in the original manuscript might be confusing for readers. I made the exposition clearer by stating the above content described in five lines explicitly. I also hope that this part has become to better clarify how the Navier-Stokes equations, the continuity equation, the Beltrami relation, pressure and external force play their roles in constructing the axisymmetric Beltrami flow.  

  1. line 165: A phrase “Equations (13) and (14) imply that” is used.
  2. lines 244-245: A sentence “If we require the solution to be nondivergent in the whole spatial region, must be real and .” was added. In passing, a phrase “solutions exist whose” is added four lines below (25).
  3. line 248: “and (4)” is added after “(23)”. The factor was recovered in (24).

Correction:  à  in equations (24) and (25).

  1. line 251: Correction: à
  2. line 284: The definition equation below (29) was changed to .
  3. lines 291-296: The sentences before and after “grossly speaking” were changed.
  4. lines 303-304: An explanation on was added in the caption of Figure 1. Labelling of the solutions are changed from A, B, C to (i), (ii) and (iii). The values of the parameters were chosen in trial and error with an expectation that solutions with a peripheral structure akin to such known exact solutions as Oseen’s and Burgers’ ones [10] exist. I suppose readers do not need to know this author’s intension.
  5. lines 311-313: Two unnecessary variables, “ ” and “ ”, were deleted. The upper value for given here is an approximate one, so I added dots after the last digit here (and hereafter).
  6. I supplemented “ ” in the first equation of (31).
  7. lines 320-325: The argument here is based on linearizing as and approximating (31) by equivalent Bessel’s differential equations. An explanation on this point is added below (31).
  8. line 353: The typo in the value of is corrected.
  9. lines 352-356: The value does not depend on because  is a function of .
  10. line 359, 361-366: “Distance” is from the symmetry axis. An explanation on the characteristics of the flow between two parallel planes expected by the solution with negative is presented in the text. For a sake of the reviewer’s reference, I present below the figures of examples of the stream lines for positive and negative A. Lower panels are for negative A and clearly show a possible candidate of flows between parallel horizontal plane.

Stream lines near the symmetry axis drawn from the numerical solutions for . Upper panel: ,Lower panel: . Left: View from obliquely above. Right: Side view. The stream for  is downward in  and upward in .

  1. line 357: The value “1” is exact while “ ” is approximate.
  2. line 443: I inserted “(necessary)” in the sentence.
  3. lines 490-491: The description below (A1) is changed as “where . Since is a function of and , so is .” Furthermore, the lower bound of the integration for  is fixed.
  4. line 490: As was described above, the definition of has been changed to avoid the ambiguity. Since I here describe the derivations of equations, not solutions of equations, there will be no need of additional explanation.
  5. lines 529-532: I changed the exposition on (C1) as is given below.

Considering that  is linear in  near  and behaves as  at large distances, we fit the solution A in Figure 1 by

(C1)

The first term on the r.h.s. is borrowed from [1] for the familiar exact solution. Choosing the parameters as , (C1) reproduces the data quite well as is shown in the upper panel of Figure 3 in the text.

* I described in lines 216-227 the steps in the process of obtaining a solution.

* In Figure 1, I adopted new labellings (i), (ii) and (iii) instead of A, B and C for the solutions.

Reviewer 2 Report

The author of the manuscript entitled "Three-dimensional unsteady axisymmetric viscous Beltrami vortex solutions to the Navier–Stokes equations" presents solutions of the Navier-Stokes to resolve Beltrami vortices, where velocity and vorticity are parallel. This involved the derivation of linearized flow equations to solve for incompressible flows. The equations are compared with the analogous Bragg-Hawthorne equation which applies to inviscid flow. The author reproduces a solution for the constant form of the coefficient of the relation between vorticity and velocity, and provides a new solution for variable form of the coefficient.

Lines 88-90: Please explain in some more detail why magnetic fields are important to Beltrami flows. I think it is from the context of the plasma flows in astrophysical models. Is this correct? If that is the case is it also applicable to other flows where plasmas are present.

Lines 290-291: I disagree with the sentence: "Grossly speaking, the width of the peak gets large with the decrease of A". The peak decreases in magnitude as the parameters A and B are reduced. It is not clear from the plot in Figure 1 which parameter contributes the most to the change in the magnitude (not the width) of the peak. Please rephrase. Also please differentiate the curve titles from the parameters.

Please check the listing of the references, it appears that the full list of references is written out for a second time see lines 630-722.

Lines 165-166: For the part of the sentence with brackets, please remove the brackets and start as a new sentence from "notice that". i.e. "only. Notice that **partial derivative condition** for regular c so that phi prime and psi are mutually dependent..."

Line 176: Please replace functional with function

Line 190: Please remove "the" from the end of the line, so that it reads "... Navier-Stokes equation (2) for axisymmetric (**partial derivative condition**) Beltrami flow ..."

Line 224-225: Please rephrase as "That this procedure is possible, is a pecularity..."

Equation (29): Please remove the error symbol and question mark that obscures the equation.

Line 369: Please correct the word "egions".

Lines 369-370: This sentence needs rephrasing to be more transparent to the general reader: "Appropriate integration regions are also understood".

Lines 512-513: Please rephrase for clarity: "in case psi and c are not spatially constant, S_r(sigma), in particular F_r when H is spatially constant, is required to be nonzero". It is not immediately clear what is meant to be nonzero - I presume it is S_r(sigma) though I could be wrong.

Author Response

Author’s reply to Reviewer_2’s comments

  1. Takahashi

Thank you for reviewing the manuscript.

The line numbers refer to the part I rewrote and have been highlighted by red fonts. The equation numbers and line numbers below are those appearing in the revised manuscript.

  1. Lines 85-91: Vorticity equation and Maxwell equation read

and

where  is the magnetic viscosity. With the correspondence of magnetic field B to vorticity , force-free motion of charged particles in magnetic field has mathematical similarity to Beltrami flow.

Yes, but the correspondence will not be restricted to the ones in astrophysics. I think strong flow may be a key point.

  1. Lines 291-292: By “width” I mean the so called “half width” or like that. The statement here will be readily understood if we notice that the parameter defines the fundamental length scale of the system (28) or (29). Note also that the width of the peak in Figure 1 has a dimension of length.
  2. In Figure 1, I adopted new labellings (i), (ii) and (iii) for the solutions.
  3. Duplication of references is deleted.

I received the reviewer’s comments given below on the quality of English language except for the item 9 which I could not get the meaning. For my responses, please see the corresponding parts in the revised manuscript.

  1. Lines 165-166: For the part of the sentence with brackets, please remove the brackets and start as a new sentence from "notice that". i.e. "only. Notice that **partial derivative condition** for regular c so that phi prime and psi are mutually dependent..."
  2. Line 177: Please replace functional with function
  3. Lines 191-192: Please remove "the" from the end of the line, so that it reads "... Navier-Stokes equation (2) for axisymmetric (**partial derivative condition**) Beltrami flow ..."
  4. Line 225: Please rephrase as "That this procedure is possible, is a peculiarity..."
  5. Equation (29): Please remove the error symbol and question mark that obscures the equation.
  6. Line 377-378: Please correct the word "egions".
  7. Lines 377-378: This sentence needs rephrasing to be more transparent to the general reader: "Appropriate integration regions are also understood".
  8. Lines 521-522: Please rephrase for clarity: "in case psi and c are not spatially constant, S_r(sigma), in particular F_r when H is spatially constant, is required to be nonzero". It is not immediately clear what is meant to be nonzero - I presume it is S_r(sigma) though I could be wrong. [Author replies: The reviewer is right.]

* Figure 1 has been redrawn.

Round 2

Reviewer 2 Report

Thank you for addressing my comments.

Just one minor correction for line 292. "gets large" should be "becomes larger".

Equation 9 has an error message in it that the author may not be able to see on their machine. See the first and third positions for g below. 

Author Response

Reply to the reviewer 2’s comment #2

Thank you for your further comment.

The words “gets large” in line 292 were changed to “becomes larger”, which are marked by blue fonts in the latest revision. 

Although I did not find any errors in Equation (29):

on my computer, I reedited the equation.

K. Takahashi
